# Bacterial Cellulose Composites with Polysaccharides Filled with Nanosized Cerium Oxide: Characterization and Cytocompatibility Assessment

**DOI:** 10.3390/polym14225001

**Published:** 2022-11-18

**Authors:** Valentina A. Petrova, Iosif V. Gofman, Alexey S. Golovkin, Alexander I. Mishanin, Natallia V. Dubashynskaya, Albert K. Khripunov, Elena M. Ivan’kova, Elena N. Vlasova, Alexandra L. Nikolaeva, Alexander E. Baranchikov, Yury A. Skorik, Alexander V. Yakimansky, Vladimir K. Ivanov

**Affiliations:** 1Institute of Macromolecular Compounds, Russian Academy of Sciences, Bolshoi VO 31, St. Petersburg 199004, Russia; 2Almazov National Medical Research Centre, Akkuratova 2, St. Petersburg 197341, Russia; 3Kurnakov Institute of General and Inorganic Chemistry, Russian Academy of Sciences, Leninskii 31, Moscow 119071, Russia

**Keywords:** nanocomposites, biopolymers, bacterial cellulose, polysaccharides, ceria nanoparticles, biomedical applications, stem cell proliferation

## Abstract

A new biocompatible nanocomposite film material for cell engineering and other biomedical applications has been prepared. It is based on the composition of natural polysaccharides filled with cerium oxide nanoparticles (CeONPs). The preparative procedure consists of successive impregnations of pressed bacterial cellulose (BC) with a sodium alginate (ALG) solution containing nanoparticles of citrate-stabilized cerium oxide and a chitosan (CS) solution. The presence of CeONPs in the polysaccharide composite matrix and the interaction of the nanoparticles with the polymer, confirmed by IR spectroscopy, change the network architecture of the composite. This leads to noticeable changes in a number of properties of the material in comparison with those of the matrix’s polysaccharide composition, viz., an increase in mechanical stiffness, a decrease in the degree of planar orientation of BC macrochains, an increase in hydrophilicity, and the shift of the processes of thermo-oxidative destruction of the material to a low-temperature region. The latter effect is considered to be caused by the redox activity of cerium oxide (reversible transitions between the states Ce^4+^ and Ce^3+^) in thermally stimulated processes in the nanocomposite films. In the equilibrium swollen state, the material retains a mechanical strength at the level of ~2 MPa. The results of in vitro tests (cultivation of multipotent mesenchymal stem cells) have demonstrated the good biocompatibility of the BC-ALG(CeONP)-CS film as cell proliferation scaffolds.

## 1. Introduction

Different polymer composites based on natural [1] and synthetic polymers [2] are used for tissue engineering. Polysaccharides are considered to be quite promising materials for creating scaffolds for tissue engineering due to their compatibility with human tissues, low toxicity, and ability to enhance regeneration processes. A number of polymer matrices for tissue engineering in various forms (films [3,4,5], different types of gels [6,7,8,9,10,11,12,13], and nonwoven materials [14,15,16,17,18,19]) have been developed using different polysaccharides. Among the great variety of polymeric materials for tissue engineering, a special place is held by bacterial cellulose (BC) [20] and BC-based composites [21].

BC, a linear, unbranched polysaccharide consisting of β-1,4-glucopyranose units, is extracellularly produced by microorganisms (*Komagataeibacter rhaeticus*) in the process of oxidative fermentation. BC consists of an ultradispersed fiber network and is characterized by a complex hierarchical morphology along with high crystallinity. The material has a unique set of properties, such as the tendency to swell in aqueous media, excellent thermal and mechanical properties, nontoxicity, and high biocompatibility. Due to its structural similarity to the components of the extracellular matrix (for example, collagen), BC is capable of taking part in complex interactions with biological tissues. This group of properties makes BC a perspective object in the development of matrices for tissue engineering.

Both BC itself [22,23,24] and different BC-based composites [25] and nanocomposites [26,27] are regarded as promising materials since one can easily control their properties in a wide range. To obtain such composites, various methods have been developed for introducing polymers into the BC and fixing them in the material (adsorption of polymers, wet molding of BC dispersed in polymer solutions, BC biosynthesis in the presence of polymers, and a combination of these methods) [28,29,30,31,32,33,34].

The modification of BC by pressing, followed by impregnation with solutions of various polyacids (alginic or hyaluronic acids or carrageenan) and a chitosan (CS) solution leads to the formation of a polyelectrolyte network. The network architecture and properties of the material change in this process. These properties depend on the type of polyacids, their structure, and the nature of their interaction with BC and CS. For example, after drying at room temperature, the composite based on sodium alginate (ALG), in contrast to the initial BC, possesses a high water-retaining capacity and porosity and demonstrates higher cytocompatibility [21]. Thus, biocompatibility, low cytotoxicity, excellent mechanical properties (especially in the swollen state), and a highly porous structure make BC and its composites a suitable material for tissue engineering applications.

In recent years, many researchers have been engrossed in studying composite biomaterials containing nanoscale components and their use in tissue repair and regeneration. For instance, experiments on the use of cerium oxide nanoparticles (CeONPs) in tissue engineering show that composites containing CeONPs as an active filler applied as scaffolds can affect the behavior of stem cells: their migration, proliferation, and differentiation [35]. The nanostructured design of a tissue implant ensures the biocompatibility of the material, providing a close resemblance to the natural extracellular matrix and facilitating the optimal implementation of the biological factors necessary for functional tissue regeneration [36].

CeONPs can stimulate tissue regenerative activity due to their oxidative potential [35]. However, despite their antioxidant activity, CeONPs can cause oxidative stress under certain conditions [37]. To increase the efficiency and reduce the toxicity of CeONPs, they are confined in polymer matrices. The proper design of polymer-CeONP composites makes it possible to modulate the antioxidant/pro-oxidant activity of nanoparticles and enhance the antimicrobial properties of the material [38]. Recent advances in the field of biomedical polymer-CeONP composites have shown how these nanoparticles augment the biomedical potential of modern polymer materials [39,40,41,42]. Being embedded into polymer scaffolds, CeONPs influence the structure and morphology of polymeric objects, thereby determining their mechanical and thermal properties, biodegradation kinetics, porosity, and swelling capacity.

The introduction of CeONPs into the BC structure by mixing the disintegrated polysaccharide gel film and the dispersion of nanoparticles causes an increase in the stiffness of the resulting nanocomposite film, along with a decrease in its ability to swell. This indicates the formation of additional intermolecular interactions in the composite via the active surface of the nanoparticles [43]. The study of this nanocomposite film as a bioresorbable scaffold for the proliferation of mesenchymal stem cells has shown a positive effect of cerium oxide on the attachment of cells to the substrate, followed by their growth.

An analysis of both the literature data and the results previously obtained by the authors convincingly confirms the importance and future prospects of the research on the multicomponent nanocomposite materials based on an already developed combination of polysaccharides. The combination includes pressed BC, as well as ALG and CS filled with CeONPs. The development and study of such composites is, undoubtedly, of scientific and practical interest.

This paper addresses the methodology of the preparation of new nanocomposites for biomedical applications with the aforesaid composition obtained by sequential impregnations of the pressed BC with ALG and CS solutions containing CeONPs. We have studied a complex of properties of the obtained materials and determined the mutual influence of the polymer and nanofiller. The cytocompatibility of the composites has also been assessed.

## 2. Materials and Methods

### 2.1. The Biosynthesis of BC

The biosynthesis of BC was carried out under surface cultivation using a B-13015 strain of *Komagataeibacter rhaeticus* (Russian National Collection of Industrial Microorganisms), according to the previously published method [44]. The obtained BC gel film was then pressed between layers of cotton fabric on an oil press until three-fold weight loss was obtained (henceforth, wrung-out BC).

### 2.2. The Citrate-Stabilized Cerium Dioxide Nanoparticles

The citrate-stabilized cerium dioxide nanoparticles used in the experiments, ~3.5–5 nm in size, were synthesized according to the protocol described in [45].

### 2.3. Preparation of Composite Films

Sodium alginate (Qingdao Bright Moon Seaweed Group Co. Ltd., Qingdao, China) with an MW = 1.3 × 10^5^ [46], chitosan (Ennagram, Pantin, France) with an MW = 1.6 × 10^5^, and the degree of deacetylation of 80% [47] were used in the fabrication of the nanocomposites.

A composite of BC–ALG–CS (control) was prepared by sequential impregnations of the partly pressed BC containing 10% of the dry substance with a 2% ALG solution and a 1% CS solution in acetic acid, according to a previously developed method [21].

When preparing BC–ALG(CeONP)–CS, the preliminarily obtained experimental data on the amount of ALG and CS solutions absorbed by pressed BC were taken into account (26 and 3%, correspondingly) [21].

To obtain a solution of ALG(CeONP), a 2% solution of ALG was mixed with a 2% citrate-stabilized dispersion of CeO_2_ under constant mechanical stirring. The ALG solution was obtained with a nanofiller content of 12% of the polymer weight. BC–ALG(CeONP)–CS was fabricated according to a previously developed technique [21] of successive impregnations of pressed BC with a 2% ALG(CeONP) solution and 1% CS solution, followed by treatment with an alcohol solution of ammonia and washing with ethyl alcohol. The resulting composite film was dried at room temperature in a fixed position. Weighing was carried out at each stage of processing to determine its composition. The composite obtained consisted of BC (71.2 wt.%), ALG (25.9 wt.%), CS (2.9 wt.%), and CeONPs (3.1% of polymer weight).

To determine the nature of the effect of the citrate-stabilized CeONP dispersion on the properties of ALG, a control film was prepared from a 2% solution of ALG containing 12% CeONPs. The solution was cast from a spinneret on a glass substrate and dried at room temperature.

### 2.4. Characterization of the Control and Composite Films

Fourier-transform infrared spectroscopy (FTIR) spectra of the ALG, ALG(CeONP), and BC–ALG(CeONP)–CS films were recorded on a Vertex 70 IR Fourier spectrometer (Bruker Optics, Ettlingen, Germany). The MIRacle ATR (Attenuated Total Reflection) reflector (Pike Technologies, Madison, Fitchburg, WI, USA) with a ZnSe working element was used in order to keep the structure of the films. When registering the ATR spectra, a correction was introduced that takes into account the penetration depth depending on the wavelength [39].

Equilibrium swelling of films—swelling after 24 h of water exposure—was determined by the gravimetric method.

The resulting films were tested via scanning electron microscopy (SEM) using a SUPRA-55VP scanning electron microscope (Carl Zeiss, Oberkochen, Germany) and via wide-angle X-ray scattering (WAXS) using a Bruker D8 DISCOVER X-ray diffractometer with CuKα (Bruker, Karlsruhe, Germany). SEM images were obtained using a secondary electron detector as well as a backscattered electron detector. To visualize the distribution of CeONPs in the samples, the films were frozen and split in liquid nitrogen, and then they were glued onto a conductive tape, sputtered by a thin layer of platinum, and tested using an energy-dispersive elemental analysis (EDX) to obtain the carts of the distribution of Ce. Element maps were collected with the help of an EDX-Max 80 mm^2^ detector (Oxford Instruments, Oxford, UK). The analysis was performed over the entire visible area of the samples.

An AG-100kNX Plus setup (Shimadzu, Osaka, Japan) operating in a uniaxial extension mode was used to study the mechanical characteristics of the films. Strip-like samples (2 × 20 mm^2^) were stretched at room temperature at a rate of 2 mm/min, according to ASTM D638 requirements. The stress–strain curves of the samples were registered during the tests. Young’s modulus (E), the break stress (σ_b_), and the ultimate deformation (ε_b_) were determined.

Thermogravimetric (TGA) and differential thermal (DTA) analyses were performed to determine the concentration of the remaining water in the films and the content of cerium oxide in the nanocomposite materials, and to characterize the impact of the nanofiller on the thermal properties of the composite films. Using TGA curves, we determined the thermal stability indices of the samples, τ_5_ and τ_10_ (the temperatures at which a polymer or a composite loses 5% and 10% of its initial weight, respectively, as a result of thermal destruction processes). A DTG-60 thermal analyzer (Shimadzu, Kyoto, Japan) was used, and the samples (~5 mg) were heated in air up to 600 °C at a rate of 5 °C/min.

### 2.5. Cultivation of Multipotent Mesenchymal Stem Cells (MMSCs)

The study of the biocompatibility of the cell culture with the scaffold was conducted on BC–ALG–CS and BC–ALG(CeONP)–CS samples. Human multipotent mesenchymal stem cells (MMSCs) obtained from subcutaneous adipose tissue of healthy donors were used as a cell culture. Cover slips of 12 mm in diameter were used as a control. The study was performed according to the Helsinki Declaration and approval was obtained from the Local Ethics Committee of the Almazov National Medical Research Centre (No. 12.26/2014; 1 December 2014). Written informed consent was obtained from all subjects prior to fat tissue biopsy.

The cell culture was performed in an alpha-MEM culture medium (Thermo Fisher Scientific, Waltham, MA, USA) supplemented with 10% fetal bovine serum, 1% L-glutamine, and 1% penicillin/streptomycin solution (Thermo Fisher Scientific, Waltham, MA, USA) in a CO_2_ incubator containing 5% CO_2_ at 37 °C. The study was performed as it was previously described elsewhere [21]. Briefly, rectangular samples of materials 12 × 8 mm^2^ in size were placed in phosphate-buffered saline (PBS) containing 2% penicillin/streptomycin solution for 30 min, and then washed three times with fresh PBS. The cover slips were treated in 70% ethanol for 10 min and then washed three times in PBS. Next, the samples and cover slips were placed in the wells of a 24-well plate. Then, 1 mL of MMSC suspension in a concentration of 50,000 cells/mL was added and cocultured for 72 h in a CO_2_ incubator containing 5% CO_2_ at 37 °C. The experiment was performed in triplicates.

After 3 days, the samples and cover slips were transferred to the wells of a new plate, washed of the residues of the nutrient medium in fresh PBS, and fixed in a 4% paraformaldehyde (PFA) solution for 10 min.

After that, samples and cover slips were washed with fresh PBS and stained with rhodamine-labeled phalloidin (Thermo Fisher Scientific, Waltham, MA, USA) in accordance with the previously developed protocol [21]. Briefly, the samples and cover slips with cells were treated with 0.05% Triton X-100 solution (Amresco, Solon, Cleveland, OH, USA) for 3 min, then washed in PBS thrice. Next, the rhodamine-labeled phalloidin solution in a 1% solution of fetal bovine serum (HyClone Laboratories, Inc., Logan, UT, USA) and PBS (1:500) was added to the wells, incubated for 20 min at room temperature in the dark, and then washed five times with PBS. Finally, cell nuclei were stained with 4,6-diamidino-2-phenylindole (DAPI, 1:40,000; Sigma-Aldrich, Co., St. Louis, MO, USA) and incubated for 40 s, and then the samples were thoroughly washed with PBS. After painting, the samples were stored in FSB in the dark at +4 °C. Cover glasses with cells from the control wells were mounted on glass slides using a preparation medium (Thermo Fisher Scientific, Waltham, WI, USA) and stored in the dark at room temperature.

Stained MMSCs were studied via fluorescence microscopy with a quantitative and qualitative analysis of adhered cells being performed. The cells were visualized using an Axio Vert inverted fluorescence microscope (Zeiss, Oberkochen, Germany) and a compatible Canon camera (Canon Europa N.V., Amstelveen, The Netherlands). Samples with cells were placed between two glass slides. DAPI fluorescence was recorded using the DAPI filter, and rhodamine-phalloidin fluorescence was recorded using the rhodamine channel. Ten different fields of view of each sample were photographed at ×100 and ×400 magnification.

The qualitative analysis consisted of assessing the morphology of MMSCs and their colonies by the stained cytoskeleton. During the quantitative analysis, the size of spheroid colonies on the sample surface was assessed. The counting of cell nuclei on the scaffold surface was not carried out due to the impossibility of accurately determining the number of nuclei in spheroids.

The obtained data were statistically analyzed using the GraphPad Prism software and the nonparametric Mann–Whitney U-test. The results are presented as the mean and the standard deviation (SD).

## 3. Results and Discussion

### 3.1. FTIR Spectra of the Studied Films

ALG, ALG(CeONP), BC–ALG–CS, and BC–ALG(CeONP)–CS films, as well as citrate-stabilized CeO_2_ powder, were studied via FTIR spectroscopy.

In the FTIR spectra of the ALG and ALG(CeONP) films (Figure 1a), one can observe bands in the range of 3400 cm^−1^ (OH stretching vibrations), 1595 cm^−1^ and 1411 cm^−1^ (asymmetric and symmetric stretching vibrations of COO^−^ groups), 1090 cm^−1^ (glycosidic bond vibrations), and 1026 cm^−1^ (C-C stretching vibrations).

When citrate-stabilized CeONPs were introduced into ALG, a shift of the 1090 and 1026 cm^−1^ bands by 3–4 cm^−1^ was observed (Figure 1b), which may presumably indicate the interaction between ALG and CeONPs.

To obtain more insight into the nature of the interactions in the system under study, a difference spectrum was obtained by subtracting the ALG matrix spectrum from the nanocomposite ALG-CeONP spectrum and comparing the resulting difference spectrum with that of citrate-stabilized CeO_2_ (Figure 1c).

In the spectrum of citrate-stabilized CeO_2_ in the region of 1800–1200 cm^−1^, there are bands at 1551 cm^−1^ and 1390 cm^−1^ (asymmetric and symmetric stretching vibrations of the COO^−^ group), as well as a shoulder in the region 1700 cm^−1^, which probably refers to the COOH vibrations of citric acid. A pronounced 1730 cm^−1^ band and the 1644 and 1440 cm^−1^ bands are noticeable in the subtraction spectrum. Such changes in the spectrum of citrate-stabilized CeO_2_ may indicate the substitution of citrate by ALG and/or the formation of the additional coordination of the COO^−^ group of ALG with cerium ions on the surface of CeONPs. The interaction between ALG and CeONPs is attributed to the presence of coordination vacancies on the surface of CeO_2_ [48,49] and the replacement of citrate with ALG can be driven by an entropy effect.

The spectra of BC–ALG–CS and BC–ALG(CeONP)–CS (Figure 2) show the following characteristic absorption bands: a wide band at 3500–3100 cm^−1^ (primary and secondary OH groups) and at 1600 cm^−1^ and 1422 cm^−1^ (asymmetric and symmetric vibrations of the COO^−^ group), and vibrations in the 1100 cm^−1^ region (glycosidic bond vibrations) and at 1029 cm^−1^ (COC stretching vibrations). When citrate-stabilized CeO_2_ was introduced into the matrix, the 1728 cm^−1^ band was also observed in the spectrum, probably due to the citric acid released. In addition, there were noticeable changes in the shape of the bands in the 1100–1000 cm^−1^ region, in the same region where band shifts were observed when comparing the spectra of ALG and ALG(CeONP) (Figure 2). It can be assumed that this fact also indicates specific interactions of CeONPs with the polysaccharide matrix.

### 3.2. Morphology of the Composite Films

Structural features of BC and both types of the BC-based composite films were studied via SEM. The BC–ALG–CS and BC–ALG(CeONP)–CS films were, additionally, tested via energy-dispersive X-ray spectroscopy (EDX). Figure 3 presents SEM images of the investigated films obtained on the films’ cryo-cleavages (Figure 3a,c,e) and on the surfaces of the same films (Figure 3b,d,f). The results of the SEM examination of BC and BC–ALG–CS are in agreement with the effects evidenced in our previous work [21]. While analyzing the SEM images of the samples’ cross-section, it is evidently seen that the films have a layered structure. In BC (Figure 3a), these layers are divided by the empty galleries of the thickness up to several microns. In both BC–ALG–CS and BC–ALG(CeONP)–CS, this interlayer space is supposed to be filled with the ALG/CS mixture (Figure 3c,e). The inner structure of the composite film filled with CeONPs (Figure 3e) seems to be denser comparing to the unfilled one, without any porosity that can be seen at the cross-section of the BC–ALG–CS film (Figure 3c).

On the surface of the initial BC film (Figure 3b), BC fibrils with diameters up to 100 nm are clearly observed. These fibrils can also be seen on the surface of the BC-ALG(CeONP)-CS film (Figure 3f), but they are partially covered by the surface layer of other polysaccharides, as the surface of the BC–ALG–CS film (Figure 3d) is fully covered by ALG/CS mixture.

The incorporation of CeONPs into the BC–ALG(CeONP)–CS film was confirmed by EDX spectra (Figure 4b).

### 3.3. X-ray Diffraction (XRD) of the Composite Films

Figure 5 shows the XRD patterns of the BC–ALG–CS control film and the corresponding composite film with CeONPs. Both of the patterns reveal two pronounced reflections characteristic of the ordered structure of BC, with intensity maxima in the 2θ regions of 14 and 22–23° [50].

When comparing the XRD patterns of the BC–ALG–CS control and BC–ALG(CeONP)–CS, one can observe a redistribution of the intensities of these two reflections. For the BC–ALG–CS sample, I_14_/I_23_ = 2.24, whereas for the BC–ALG(CeONP)–CS composite, I_14_/I_22_ = 1.36. This effect is attributed to a change in the degree of planar orientation of BC macrochains in the sample: in BC–ALG(CeONP)–CS, the planar orientation factor is lower than that in the control BC–ALG–CS film.

In addition to the BC reflections, the XRD pattern of BC–ALG(CeONP)–CS exhibits weak reflections in the regions of 2θ = 28.7°, 33°, 47.5°, and 56.3°, which correspond to reflections from the CeO_2_ crystal lattice planes (111), (200), (220), and (311), respectively (the cubic fluorite crystal structure: ICDD PDF card #34-394, data from NIST—National Institute of Standards and Technology, Gaithersburg, MD, USA). When evaluating the intensity of these reflections, one should take into account the fact that for CeONPs in some polymer matrices, the results depend upon the concentration of the filler and the character of the ceria–polymer interaction [43,51,52,53]. The weak intensity of these reflections in the XRD pattern of the studied nanocomposite may indicate the interaction of ALG with CeONPs.

### 3.4. Swelling of Composite Films in Water

As was demonstrated in [21], the BC–ALG–CS composite, in contrast to the BC sample dried under similar conditions, retained a high degree of swelling in water (Table 1). The introduction of CeONPs into the composition led to an almost two-fold increase in the hydrophilicity of the material. This is most likely to occur as a result of a change in the structure of the ALG layer, due to the interaction between ALG and citrate-stabilized CeONPs.

### 3.5. Mechanical Properties of the Composite Films

Data on the mechanical properties of the film composite materials are given in Table 2, with the stress–strain curves being presented in Figure 6.

Like many polysaccharide films, the examined materials in the dry state are characterized by a high stiffness (elastic modulus 6–8 GPa) combined with a low ultimate deformation (3.6–4.9%). However, such a deformation extent is sufficient for the successful use of the studied films in a number of biomedical applications. Moreover, all the materials possess high-strength characteristics, which are associated precisely with their high stiffness (high elastic modulus, E).

The stiffness of the BC–ALG(CeONP)–CS film exceeds that of the control materials with no CeONPs (the elastic modulus of the composite film is 1.15 times higher than that of the BC–ALG–CS sample). Such an increase in the elastic modulus caused by the introduction of cerium oxide into the polysaccharide matrix clearly indicates the formation of a system of additional intermolecular interactions in BC–ALG(CeONP)–CS (through the surface of the nanoparticles) in addition to those existing in the control polysaccharide material.

Since the films under study were supposed to be used as scaffolds in tissue engineering in a swollen (in aqueous solutions) state, an additional experiment was carried out to determine the mechanical characteristics of the BC–ALG(CeONP)–CS film in an equilibrium swollen state after being stored in water. The storing was performed in the same conditions as those in which moisture absorption was determined (Table 2, Figure 6). It was shown that, even in this case, the BC–ALG(CeONP)–CS composite kept its strength and ultimate deformation at the level of ~2 MPa and 4%, respectively, which is enough for its practical use.

### 3.6. Thermal Analysis of the Composite Films

To ascertain the concentration of cerium oxide in BC–ALG(CeONP)–CS and evaluate its effect on the thermal behavior of the material, a simultaneous thermal analysis of the films (TGA + DTA) was performed.

#### 3.6.1. TGA of the Composite Films

According to TGA data, the thermo-oxidative destruction of all studied films begins in the temperature range of 200–300 °C and ends at 500 °C (Figure 7). When the latter is reached, polysaccharide samples (BC and BC–ALG–CS) are completely degraded, and the degradation products convert to the gas phase (the mass of the sample drops down to ~0). It is worth mentioning that the mass of the BC–ALG(CeONP)–CS sample decreases to 3.6% of the initial value and remains constant upon further heating. After the subtraction of the loss attributed to the evaporation of volatile impurities (5.4%, Table 3) from the total mass loss, we obtain the mass fraction of CeO_2_ in the “dry” material that equals to 3.8%.

When volatile impurities (3.5–5.5%, Table 3) are eliminated from the samples, the thermal destruction of the films proceeds in two stages, which is typical of a number of polysaccharides. The first (low-temperature) stage occurs in the temperature range of 170–360 °C in the BC–ALG(CeONP)–CS film and at 190–380 °C in the control one. The decomposition processes at these temperatures correspond to the initial stage of pyrolysis, including simultaneous dehydration, depolymerization, and decomposition of monomeric units of polysaccharides. Pyrolysis causes random cleavage of glycosidic bonds, followed by further decomposition resulting in C2-, C3-, and C6-fatty acid formation, including acetic acid and butyric acid. The processes of polymer degradation are completed at the second (high-temperature) stage in the region of 490–500 °C. At this stage, the oligomeric and monomeric products formed at the first stage degrade in an oxygen-containing atmosphere, bringing about the formation of gaseous substances [54,55,56,57]. A similar pattern of thermo-oxidative degradation of BC was reported in our previous studies [43].

A comparative analysis of the thermal characteristics of BC–ALG(CeONP)–CS and the control films reveals a pronounced shift to the lower values in the temperature of the thermo-oxidative degradation of the material (τ_5_ and τ_10_ drop down by 45–50 °C). This is supposed to result from the introduction of CeONPs into the polysaccharide matrix.

It should be noted that the thermal degradation characteristics of the BC–ALG–CS control film are quite similar to those of the single-component BC film.

#### 3.6.2. DTG and DTA of the Composite Films

The same temperature shift in the maximum thermal degradation rate at both stages is also observed on the DTG and DTA curves of the BC–ALG(CeONP)–CS sample (Figure 8a,b). This result agrees well with the data on the catalytic effect of cerium oxide on the destruction processes in organic compounds.

### 3.7. Cultivation of Multipotent Mesenchymal Stem Cells (MMSCs)

The cells on the cover glasses were evenly spaced and spread out on the surface of the glass with the formation of a confluent/sub-confluent monolayer, and had a typical elongated shape with multiple outgrowths and clearly detected actin microfilaments. Some of the MMSCs were in the process of division (Figure 9, Figure 10 and Figure 11). In the case of the BC–ALG–CS, the cells were located evenly on the surface, mainly in the form of a monolayer with a confluence up to 40% with single spheroids. Most of the cells had a typical elongated shape with longitudinal striation and multiple processes, and outgrowths through which they were connected to neighboring cells. A smaller part of the cells on the sample surface had a rounded or close to rounded shape without processes. Spheroids were single with signs of cell migration along the periphery of colonies (Figure 9, Figure 10 and Figure 11).

In the case of BC–ALG(CeONP)–CS, cells were located superficially and nonuniformly on the scaffold surface in the form of flat colonies, spheroids, and individual cells. In flat colonies, MMSCs had a typical elongated shape with a longitudinal striation more pronounced than in the BC–ALG–CS group and multiple processes. Some cells were in the process of division. The cell density in colonies varied from 30–40 to 80–90% in different areas. Spheroid colonies were detected in small quantities (more than in the BC–ALG–CS group) with signs of active cell migration along the periphery (more pronounced compared to the BC–ALG–CS group). Some of the spheroids were visualized in flat colonies and merged with them due to the pronounced migration of cells from the spheroid colonies. At the same time, there was no statistically significant difference in the maximum longitudinal size of spheroid colonies in both groups of samples (*p* > 0.05) (Table 4). Separately located cells were single, and had a rounded or close to rounded shape without appendages (Figure 9, Figure 10 and Figure 11).

The BC-ALG(CeONP)-CS composite film was shown to have better biocompatibility as compared to BC-ALG-CS. This effect can be caused by the complex action of different factors for different reasons. In analyzing these reasons, one should take into account not only the biological activity of the CeONPs, but also the possible action of the nanosized filler upon the physical properties of the matrix material. It is well known that the adhesion of cells to the scaffold surface is strongly affected by the micro-relief and texture of the latter [58]. The introduction of CeONPs into the polysaccharide matrix is supposed to create a surface structure that is most suitable for the formation of contacts of the cell with the scaffold (see Figure 3f). A certain electric charge on the scaffold surface also ensures optimal conditions for proliferation [3]. In the case of the nanocomposite material formed in our work, this additional electric charge might be provided by the nanoparticles [59].

A number of studies [34] have found that spheroidal MMSC colonies are formed on the surface of BC matrices. This organization of cell colonies (as compared to the two-dimensional monolayer culture) has a number of features concerning both the process of the formation of aggregates of multicellular spheroids and the properties of the cells composing these spheroids. MMSC spheroids have a better survival compared to single-cell suspensions in vivo, but they have a smaller survival advantage than MMSCs when grown under normal conditions of a 2D culture in vitro [60]. In living tissue, cells exist in a 3D microenvironment with complex interactions between cells and matrix and with complex dynamics of nutrient and cellular transport [61,62]. The developed BC-based material containing CeONPs showed some advantages over the control when culturing MMSCs in vitro, which makes this material promising for in vivo applications.

## 4. Conclusions

A new biocompatible polymeric nanocomposite material for cell engineering and other biomedical applications has been developed and studied. The material, based on bacterial cellulose in combination with sodium alginate and chitosan, contains cerium oxide nanoparticles. For the preparation of the films of the said composition, a step-by-step protocol was optimized, including successive impregnations of pre-pressed bacterial cellulose with a sodium alginate solution containing ceria nanoparticles and with a chitosan solution.

The interaction of the active filler nanoparticles with the polysaccharide matrix in the resulting composite material was corroborated by spectral studies. The results of a comprehensive study of the characteristics of the obtained composite material confirm the effect of nanosized cerium oxide on the properties of polysaccharide films. It is shown that the introduction of the nanoparticles into the polymer composite changes the structure of the material. For instance, the redistribution of the intensities of the two main BC reflections on the diffraction pattern of the nanocomposite reveals a change in the degree of planar orientation of cellulose macrochains.

A change in the network architecture of the polysaccharide composition caused by the introduction of a nanofiller affects the morphology of the composite: according to SEM, the BC-ALG(CeONP)-CS cryo-cleavage is characterized by a denser layered structure compared to the control BC-ALG-CS sample.

Nanocomposite films possess increased hydrophilicity and are characterized by a high elasticity modulus while retaining significant strength, both in dry and swollen states.

The BC-ALG(CeONP)-CS nanocomposite used as a substrate for cell engineering demonstrates an increased intensity in the cultivation process of multipotent mesenchymal stem cells in in vitro tests compared to the control BC-ALG-CS sample.

The studies performed have shown that the design of the nanocomposite, based on the intermolecular interactions of polysaccharides and the nanofiller, ensures the biocompatibility of the material and a close similarity to the natural extracellular matrix. The prospects of using the developed nanocomposite as a material for biomedical purposes are shown. The material can be used for tissue repair and regeneration provided further investigations in this direction are carried out.

## Figures and Tables

**Figure 1 polymers-14-05001-f001:**
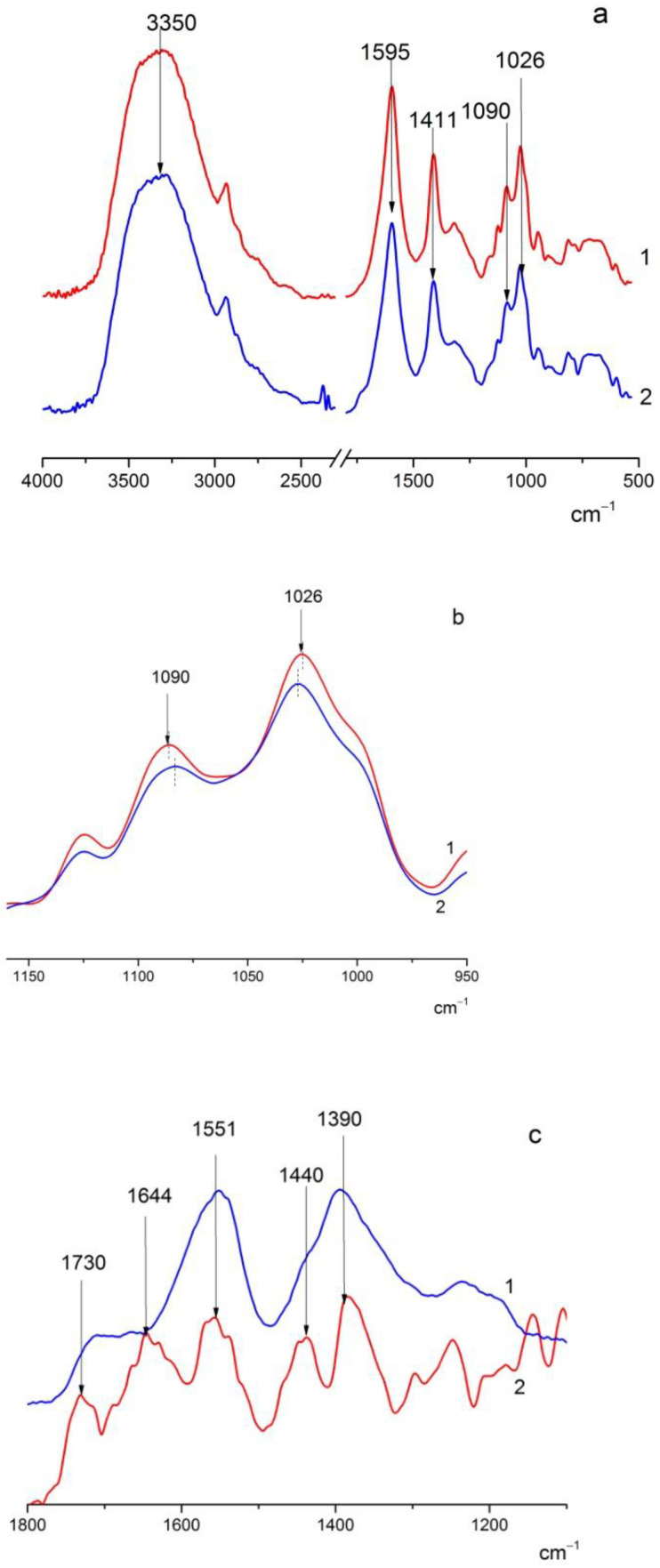
FTIR spectra of the studied films. (**a**): (1) sodium alginate (ALG) and (2) ALG-ceria nanoparticles (CeONPs); (**b**): fragment of spectra 1a on enlarged scale, ALG (1) and ALG(CeONP) (2); (**c**): citrate-stabilized CeO_2_ (1) and difference spectrum obtained by subtracting the spectrum of ALG from that of the ALG(CeONP) composite (2).

**Figure 2 polymers-14-05001-f002:**
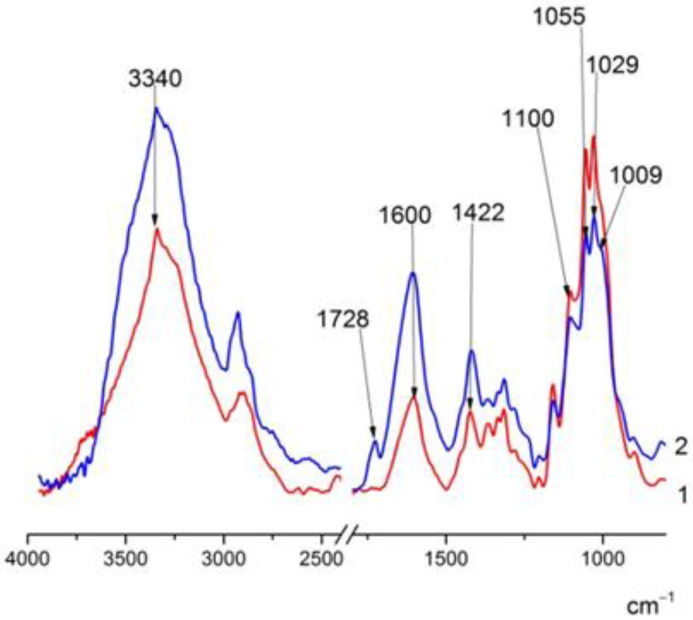
FTIR spectra: (1) BC–ALG–chitosan (CS_), (2) BC–ALG(CeONP)–CS.

**Figure 3 polymers-14-05001-f003:**
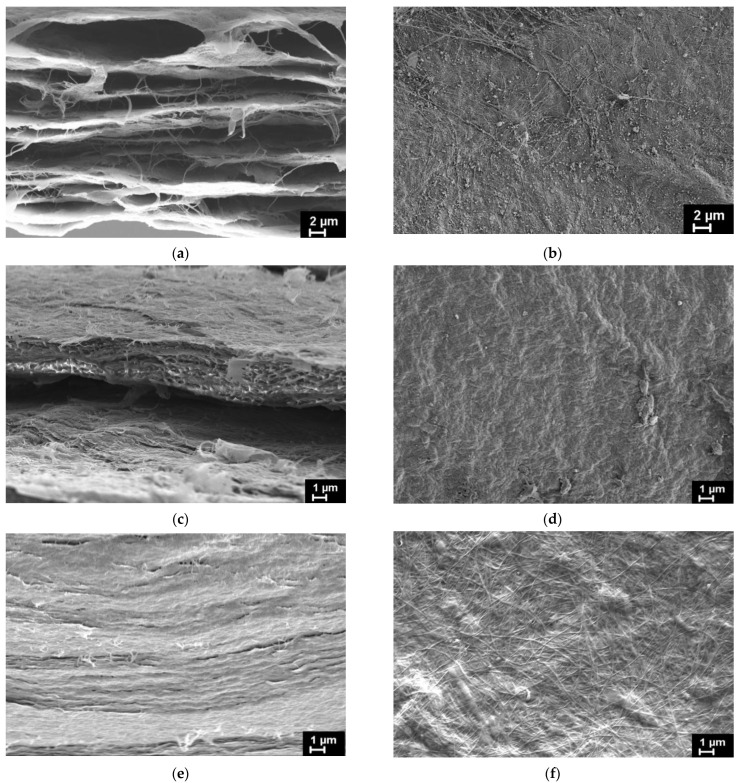
SEM images of BC film (**a**,**b**), BC–ALG–CS film (**c**,**d**), and BC–ALG(CeONP)–CS film (**e**,**f**): (**a**,**c**,**e**)—cryo-cleaved surfaces; (**b**,**d**,**f**)—film surfaces.

**Figure 4 polymers-14-05001-f004:**
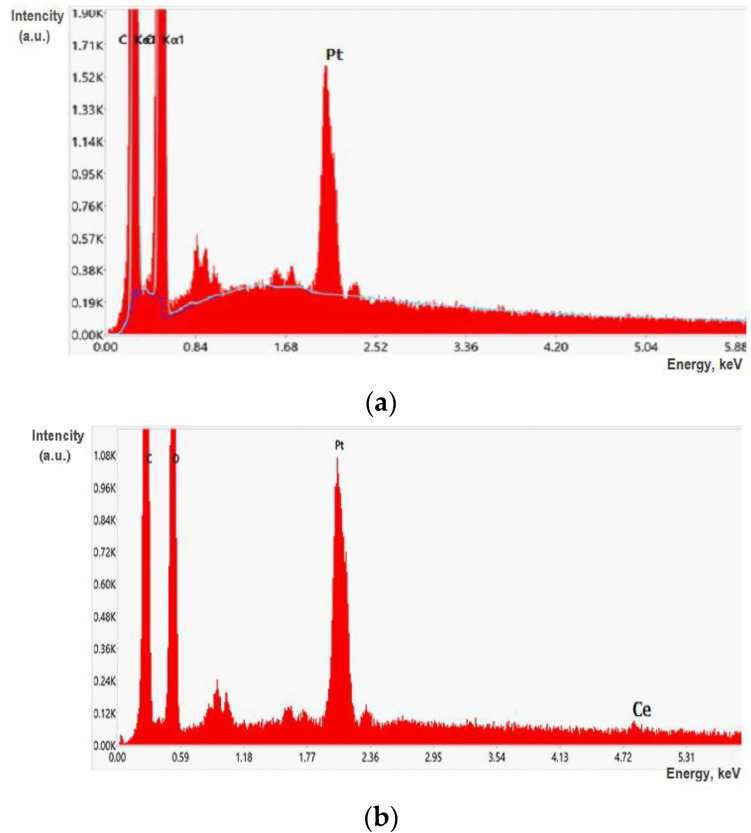
EDX spectra registered on the BC–ALG–CS film (**a**) and the BC–ALG(CeONP)–CS film (**b**).

**Figure 5 polymers-14-05001-f005:**
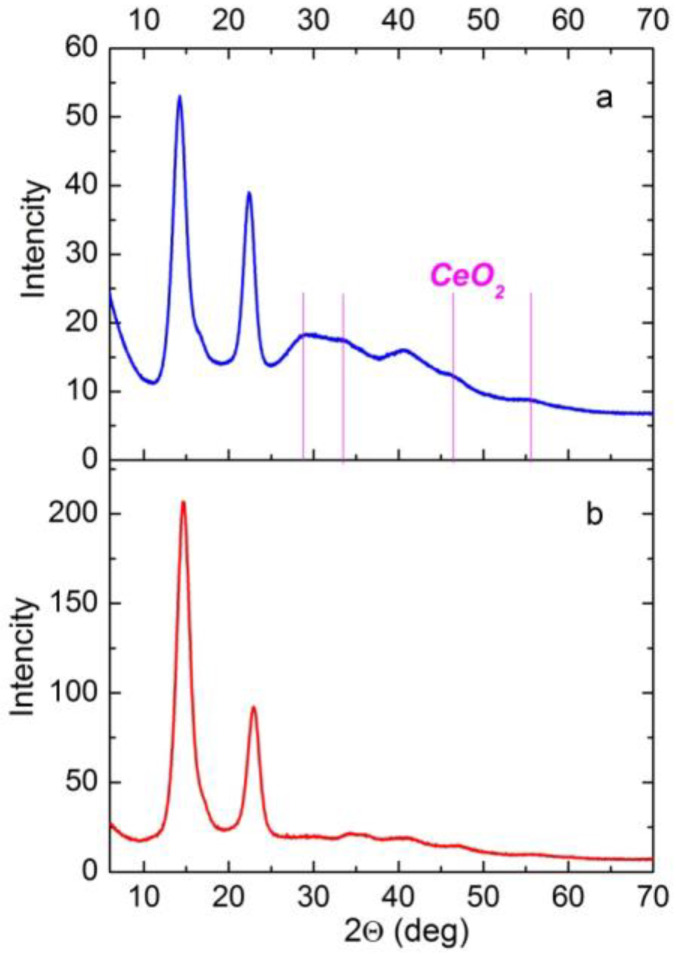
XRD patterns of BC–ALG(CeONP)–CS (**a**) and BC–ALG–CS (**b**).

**Figure 6 polymers-14-05001-f006:**
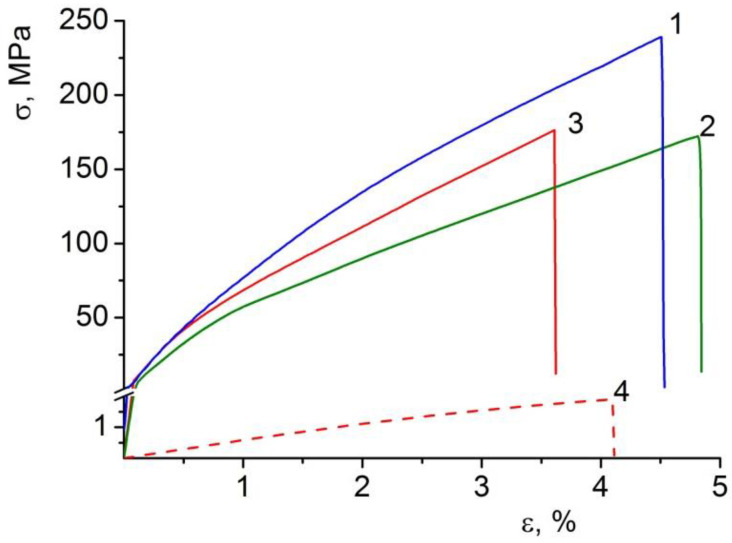
Stress–strain curves of (1) BC, (2) BC–ALG–CS, and (3) BC–ALG(CeONP)–CS dry films, and (4) BC–ALG(CeONP)–CS film swollen in water.

**Figure 7 polymers-14-05001-f007:**
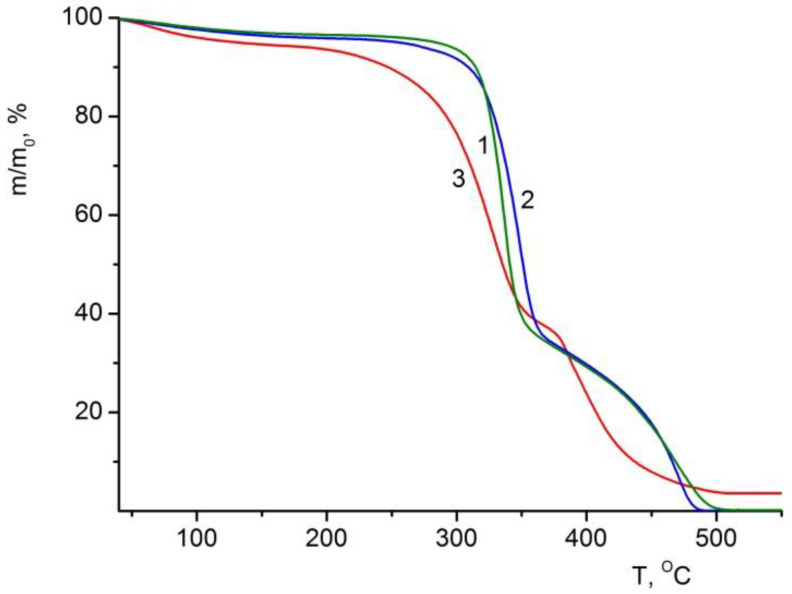
TGA curves of (1) BC, (2) BC–ALG–CS, and (3) BC–ALG(CeONP)–CS films.

**Figure 8 polymers-14-05001-f008:**
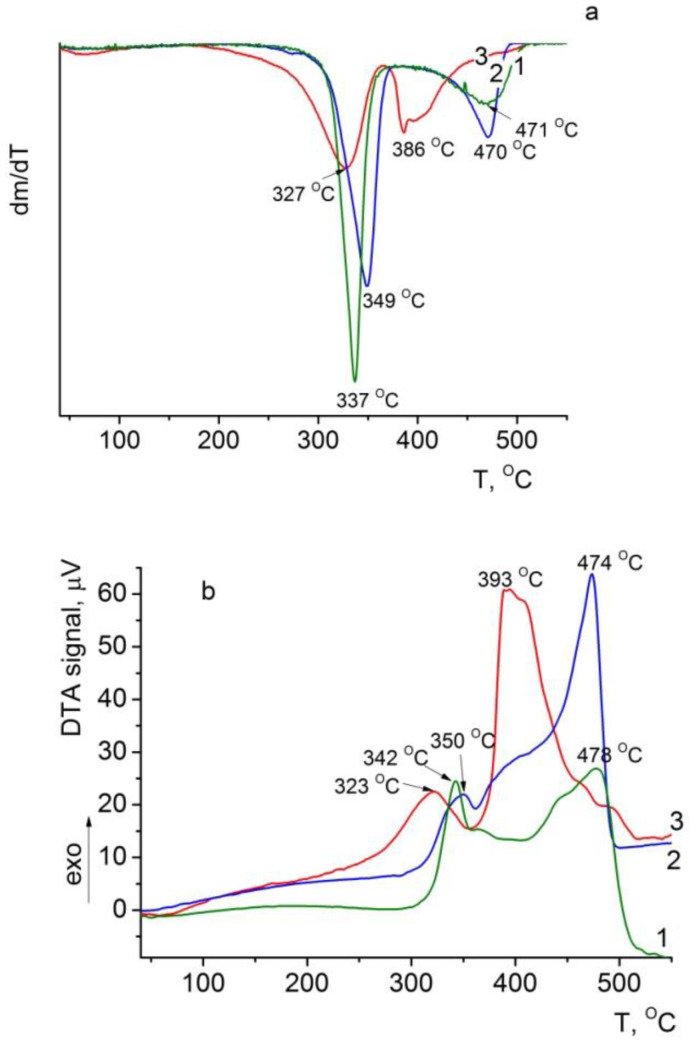
DTG (**a**) and DTA (**b**) curves of (1) BC, (2) BC–ALG–CS, and (3) BC–ALG(CeONP)–CS films.

**Figure 9 polymers-14-05001-f009:**
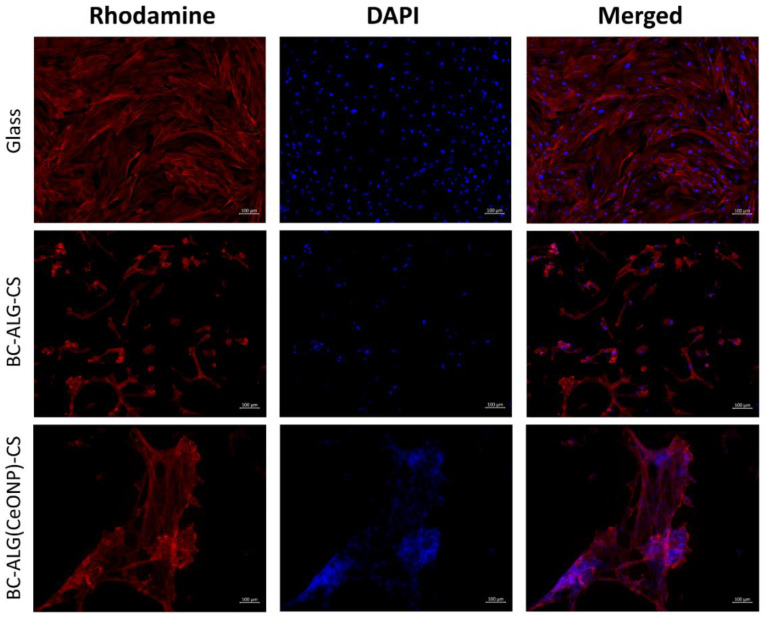
Multipotent mesenchymal stem cells adhered to the surfaces of glass and scaffolds. Fibrillar actin of the cytoskeleton was stained with rhodamine fluorochrome; nuclei were stained with DAPI. Combined two-channel image, magnification ×100.

**Figure 10 polymers-14-05001-f010:**
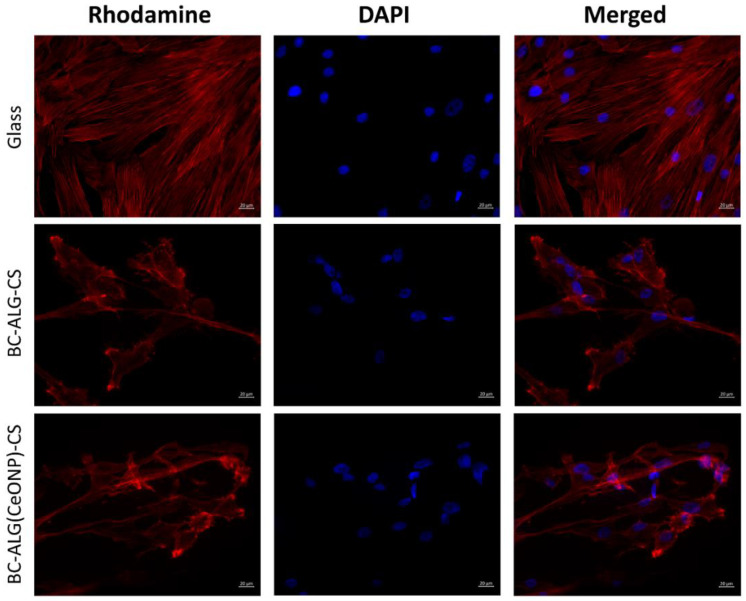
Multipotent mesenchymal stem cells adhered to the surfaces of glass and scaffolds. Fibrillar actin of the cytoskeleton was stained with rhodamine fluorochrome; nuclei were stained with DAPI. Combined two-channel image, magnification ×400.

**Figure 11 polymers-14-05001-f011:**
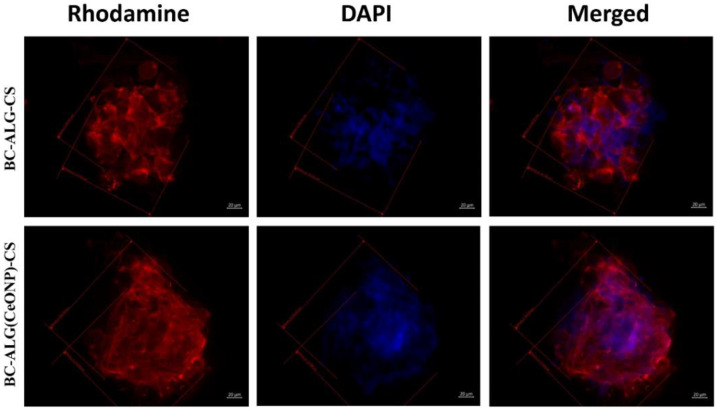
Multipotent mesenchymal stem cell spheroids formed on different scaffolds on the third day after seeding. Fibrillar actin of the cytoskeleton was stained with rhodamine fluorochrome; nuclei were stained with DAPI. Combined two-channel image, magnification ×400.

**Table 1 polymers-14-05001-t001:** Equilibrium swelling of bacterial cellulose (BC), BC–ALG–CS, and BC–ALG(CeONP)–CS films.

Sample	Swelling in Water (g/g)
Bacterial cellulose (BC)	1.9
BC–sodium alginate (ALG)–chitosan (CS)	24.0
BC–ALG-cerium oxide nanoparticles (CeONP)–CS	42.0

**Table 2 polymers-14-05001-t002:** Mechanical properties of BC, BC–ALG–CS, and BC–ALG(CeONP)–CS films.

Sample	E	σ_b_ (MPa)	ε_b_ (%)
BC dry	6.07 ± 0.18 GPa	236 ± 11	4.7 ± 0.6
BC–ALG–CS dry	7.13 ± 0.08 GPa	172 ± 9	4.9 ± 0.2
BC–ALG(CeONP)–CS dry	8.21 ± 0.53 GPa	176 ± 7	3.6 ± 0.4
BC–ALG(CeONP)–CS swollen in water	39 ± 2 MPa	1.9 ± 0.2	4.1 ± 0.3

**Table 3 polymers-14-05001-t003:** Thermal stability indices of BC, BC–ALG–CS, and BC–ALG(CeONP)–CS films in air atmosphere.

Sample	Volatile Impurities, %	Thermal Stability Indices (°C)	
		τ_5_	τ_10_
BC	3.5	310	319
BC–ALG–CS	4.1	304	320
BC–ALG(CeONP)–CS	5.4	251	276

**Table 4 polymers-14-05001-t004:** Characteristics of multipotent mesenchymal stem cells and cell colonies formed on the surface of the scaffolds on the third day after seeding.

Sample	Nonadhered Cells	Type of Colonies	Maximal Longitudinal Site of Spheroids (µm)	Cell Migration from Spheroid
Glass	–	monolayer		
BC–ALG–CS	multiple	monolayer+ single spheroids	154 ± 2	+
BC–ALG(CeONP)–CS	single	monolayer+ spheroids	163 ± 5	++

+ and ++ denote the degree of cell migration from spheroid.

## Data Availability

Data available upon request.

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
