# Peer review of "Bacterial Cellulose Composites with Polysaccharides Filled with Nanosized Cerium Oxide: Characterization and Cytocompatibility Assessment"

_polymers, 2022, doi:10.3390/polym14225001_

Round 1

Reviewer 1 Report

The authors have done a great job presenting a new nanocomposite biomaterial using cerium oxide nanoparticles with bacterial cellulose, alginate and chitosan hydrogels substrates as a novel nanocomposite. There are some concerns and comments which must be addressed however before this manuscript can be considered for publication:

1. The manuscript must include multipanel final figure for instance : Figures 1,2, and 3 which all show FTIR data can be clubbed as  Figure 1 parts a, b, and c. Please make sure that the manuscript does not include unnecessary single figures but more informative multipanel figures with detailed figure captions.

2. In Figure 5 all panels of SEM images must be individually labeled within each image for clarity. The figure captions must also include key inferences drawn from the SEM imaging. Magnifications must be clearly listed for clarity to the readers.

3. In figure 6 the EDX data must have a Y-axis. The Y-axis appears tp be cut off and is not clearly visible. Change the orientation of parts a and b to verticle for clarity.

4. The TGA data must be clearly labeled as parts a,b, and c with corresponding headings associated with each graph.

5. The font sizes of the text within all figures must be consistent and comparable to the text within the manuscript.

6. In figure 10, the rhodamine DAPi staining images show very poor cell spreading observed on the BC-ALG-CS and the BC-ALG(CeONP)-CS scaffolds. How do the authors explain this? How is this material more beneficial to the cells as compared t conventional hydrogels that allow improved cell spreading?

7. The study should analyze other key components of cell behavior such as cell proliferation, viability, cytotoxicity, and apoptosis response to quantitatively evaluate the cellular response to these novel scaffolds.

8. The study should also include some PCR data to understand how gene expression changes over time upon exposure to these novel nanocomposite scaffolds.

Author Response

Answers to the notes of the Reviewer#1

The authors are sincerely grateful to the reviewer for the critical review and valuable comments. Here are our responses; major changes in the text are highlighted in red. We hope that our efforts to improve the paper will be sufficient for its acceptance.

  1. The manuscript must include multipanel final figure for instance : Figures 1,2, and 3 which all show FTIR data can be clubbed as  Figure 1 parts a, b, and c. Please make sure that the manuscript does not include unnecessary single figures but more informative multipanel figures with detailed figure captions.

Single figures have been combined into panels, where possible. The figure captions were revised.

  1. In Figure 5 all panels of SEM images must be individually labeled within each image for clarity. The figure captions must also include key inferences drawn from the SEM imaging. Magnifications must be clearly listed for clarity to the readers.

SEM images on Figure 5 (Figure 3 in the revised version) have been individually labeled.

  1. In figure 6 the EDX data must have a Y-axis. The Y-axis appears tp be cut off and is not clearly visible. Change the orientation of parts a and b to verticle for clarity.

Figure 6 has been corrected.

  1. The TGA data must be clearly labeled as parts a,b, and c with corresponding headings associated with each graph.

For the clarity, Section 3.6. (Thermal analysis of the composite films) has been divided into two sub-sections. Figures 7 and 8 have also been modified.

  1. The font sizes of the text within all figures must be consistent and comparable to the text within the manuscript.

This has been corrected.

  1. In figure 10, the rhodamine DAPi staining images show very poor cell spreading observed on the BC-ALG-CS and the BC-ALG(CeONP)-CS scaffolds. How do the authors explain this? How is this material more beneficial to the cells as compared t conventional hydrogels that allow improved cell spreading?

The formation of cell monolayer on a surface of culture plate is one of the main properties of mesenchyme stem cells (MSCs). MSCs adhere and proliferate while spreading and specifically changing their morphology. These properties are very specific for MSCs and this is one of the reasons why their cultivation on culture plate is used as a control of cell quality. Besides, cultivation on culture plate is known to be the optimal condition for these cells. Thus, cell morphology, proliferation level etc. are used as reference to compare with any other new materials and to perform the decision about biocompatible properties of these materials. That is why we used cover slips as a reference material and cultivation on it, as a control of cell culture.

Bacterial cellulose composites are very porous materials with intricate surface topography. Obviously this topography do not looks like optimal for cell adhesion and proliferation that proved by number of spheroids and individual cells presented on the surface. Cells were spread and elongated along the fibers of the material. That is normal for adherent cell cultures. But because of the topography they were not able to form equal monolayer as it was performed on cover slips. Thus, we came to the conclusion about satisfactory biocompatibility properties of the studied bacterial cellulose composites.

In living tissue, cells exist in a 3D microenvironment with complex interactions between cells and matrix and complex dynamics of nutrient and cell transport. Various hydrogel matrices are most similar to the structure of the natural extracellular matrix. Unlike other hydrogel structures, the developed composite is formed only through intermolecular interaction, without the use of crosslinking agents. In the swollen state the composite, due to the designed structure, preserves the BC architecture and represents a strong and stable gel film, which also features it in comparison with other hydrogel composites. The developed matrix material comprising cerium oxide showed some advantages as compared to the control when culturing the MMSCs in vitro, which makes this material promising for in vivo use.

  1. The study should analyze other key components of cell behavior such as cell proliferation, viability, cytotoxicity, and apoptosis response to quantitatively evaluate the cellular response to these novel scaffolds.

Actually, cytotoxicity testing, investigation of biocompatibility properties are the important characteristics to study for any new materials. All in vitro tests as well as in vivo tests have to be performed before development medical and/or tissue engineering devices and translation them into the medicine. In this study we present primary results of in vitro testing of the novel materials. These tests can be regarded as first level testing declaring properties and abilities of these materials. 

  1. The study should also include some PCR data to understand how gene expression changes over time upon exposure to these novel nanocomposite scaffolds.

We agree with the reviewer that experiments including PCR analysis of gene expression are necessary on the next stage of testing. Gene expression analysis can demonstrate fine-tune interactions between seeded cells and materials. Understanding of these interactions is valuable for further development of tissue engineering scaffolds and devices. We are planning to perform such experiments but now we are limited for some points.

Reviewer 2 Report

The paper presented by Petrova  et al. described the Bacterial Cellulose Composites with Polysaccharides Filled with Nanosized Cerium Oxide. The authors are suggested to go through the following comments and revise the manuscript accordingly:

-Why authors was chose CeO NPs as filler?

- There are many studies investigating the importance of the topic , Please add these references to your introduction and discussion parts of the manuscript and compare and bold your study novelty: https://doi.org/10.1016/j.jddst.2020.101916, https://doi.org/10.3390/pharmaceutics12080725, https://doi.org/10.1002/advs.202003535, https://doi.org/10.1016/j.jconrel.2022.05.062, https://doi.org/10.3390/polym14061259

-Figure 4 didn’t present in the paper!

-what is the size of nanofibers in SEM image? Please add a size

- It is better to provide a graphical abstract to present main goal of paper

- The conclusions of the work are not well established and are not solid

- There are some spelling errors and logic problems in the text that need attention. Moreover, the typos in the manuscript need to be double-checked. For example, Line 126

Author Response

Answers to the notes of the Reviewer#2

The authors are sincerely grateful to the reviewer for the critical review and valuable comments. Here are our responses; major changes in the text are highlighted in red. We hope that our efforts to improve the paper will be sufficient for its acceptance.

-Why authors was chose CeO NPs as filler?

The reasons of the decision to use just the ceria nanoparticles as the active filler of the polysaccharides compositions were described in the introduction of our manuscript (page 2). Following the recommendation of the Reviewer the appropriate part of the introduction was additionally enlarged. The main reason of our choice of the nanofiller type is the remarkable set of properties of nano-ceria. For instance, experiments on the use of cerium oxide nanoparticles (CeONPs) in tissue engineering show that composites containing CeONPs as active filler applied as scaffolds can affect the behavior of stem cells: their migration, proliferation, and differentiation [35]. The nanostructured design of a tissue implant ensures the biocompatibility of the material, providing a close resemblance to the natural extracellular matrix and facilitating the optimal implementation of the biological factors necessary for functional tissue regeneration [36].

CeONPs can stimulate tissue regenerative activity due to their oxidative potential [35]. However, despite their antioxidant activity, CeONPs can cause oxidative stress under certain conditions [37]. To increase the efficiency and reduce the toxicity of CeONPs, they are confined in polymer matrices. The proper design of polymer-CeONP composites makes it possible to modulate the antioxidant/prooxidant activity of nanoparticles and enhance the antimicrobial properties of the material [38]. Recent advances in the field of biomedical polymer-CeONP composites have shown how these nanoparticles augment the biomedical potential of modern polymer materials [39-42]. Being embedded into polymer scaffolds, CeONPs influence the structure and morphology of polymeric objects, thereby determining their mechanical and thermal properties, biodegradation kinetics, porosity, and swelling capacity.

- There are many studies investigating the importance of the topic , Please add these references to your introduction and discussion parts of the manuscript and compare and bold your study novelty: https://doi.org/10.1016/j.jddst.2020.101916, https://doi.org/10.3390/pharmaceutics12080725, https://doi.org/10.1002/advs.202003535, https://doi.org/10.1016/j.jconrel.2022.05.062

The Introduction section has been expanded, and some of the recommended references have been included in the text.

-Figure 4 didn’t present in the paper!

This figure (Figure 4. FTIR spectra: (1) BC–ALG–chitosan (CS_), (2) BC–ALG(CeONP)–CS) was presented in the page 8 of the manuscript and described in the page 7. Now the figures numbering has been corrected throughout the manuscript as recommended the Reviewer # 1. In the revised text it can be found as the Figure 2 (page 7).

-what is the size of nanofibers in SEM image? Please add a size

The diameter of the BC nanofibers is varied in a range of 20-100 nm. The appropriate information was introduced into the text of section 3.2. Morphology of the composite films. For clarity, we have added the clearly visible scale bars to all the SEM images.

- It is better to provide a graphical abstract to present main goal of paper

The graphical abstract has been prepared and uploaded through the MDPI submission system.

- The conclusions of the work are not well established and are not solid.

We have added some discussion and conclusions to Section 3.7. Cultivation of multipotent mesenchymal stem cells (MMSC).

- There are some spelling errors and logic problems in the text that need attention. Moreover, the typos in the manuscript need to be double-checked. For example, Line 126

We have tried our best to correct spelling errors and hope that additional corrections will be made by the English editor during the proofreading stage of the paper processing.

Round 2

Reviewer 2 Report

-